# Real-World Data Regarding Dalbavancin Use before and during the COVID-19 Pandemic—A Single-Center Retrospective Study

**DOI:** 10.3390/antibiotics12071205

**Published:** 2023-07-19

**Authors:** Petros Ioannou, Nora Wolff, Anna Mathioudaki, Christos Spanias, Nikolaos Spernovasilis, Diamantis P. Kofteridis

**Affiliations:** 1School of Medicine, University of Crete, 71003 Heraklion, Greece; 2Internal Medicine Department, University Hospital of Heraklion, 71110 Heraklion, Greece; 3Department of Pharmacy, University Hospital of Heraklion, 71110 Heraklion, Greece; 4German Oncology Center, Limassol 4108, Cyprus

**Keywords:** *Staphylococcus*, *Enterococcus*, dalbavancin, methicillin-resistant *Staphylococcus aureus*, MRSA, endocarditis, skin and soft tissue infection, osteomyelitis, septic arthritis

## Abstract

Dalbavancin is a semisynthetic lipoglycopeptide, which possesses great potential for bactericidal activity similar to antimicrobials with the same mechanism of action, such as vancomycin and teicoplanin. Due to its very prolonged half-life, it can be used in a single or two-dose regimen to treat infections by Gram-positive microorganisms, even resistant ones, such as methicillin-resistant *Staphylococcus aureus* (MRSA). Currently, it is approved only for the treatment of acute bacterial skin and skin structure infections (ABSSSIs). The aim of this study was to investigate the clinical and microbiological characteristics of patients to whom dalbavancin was administered at the University Hospital of Heraklion and evaluate its use in regard to the COVID-19 pandemic. In total, 146 patients were included in this retrospective cohort study evaluating the use of dalbavancin from the first time it was used in 2017 until the end of 2022. The median age was 68 years (range: 21–96 years), and 86 (59%) patients were male. The most common indications for dalbavancin use were osteoarticular infections in 43%, followed by ABSSSIs in 37%, and cardiovascular infections in 10%. Dalbavancin was used empirically in one out of three patients, most commonly with the indication of ABSSSIs, and most commonly in the post-COVID-19 era. The most frequently isolated pathogens were coagulase-negative staphylococci in 70%, *S. aureus* in 27%, *Enterococcus* spp. in 22%, and *Streptococcus* spp. in 8%, while one out of three infections were polymicrobial. In 12% of patients, the infection was not cured, but no patients died. For patients with ABSSSIs, endocarditis and vascular infections, and bacteremia, the cure rates were more than 90%, and in osteoarticular infections, the cure rate was 76%. Thus, dalbavancin has great potential for use in complicated and invasive infections that may require prolonged intravenous antimicrobial treatment. However, further studies are required to formally investigate its role in such infections.

## 1. Introduction

The development of glycopeptides such as vancomycin or teicoplanin has allowed adequate treatment of Gram-positive infections such as MRSA or ampicillin-resistant *Enterococcus* [1]. Dalbavancin is a semisynthetic lipoglycopeptide, which possesses a great potential for bactericidal activity similar to antimicrobials with the same mechanism of action, such as vancomycin and teicoplanin [2]. Similar to other lipoglycopeptides, the mechanism of action of dalbavancin includes targeting the C-terminal subunit of peptidoglycan precursors. Dalbavancin has a long terminal half-life ranging from 149 to 250 h, a property that allows its use at a pace of a once-weekly or biweekly dose, therefore making it suitable for outpatient parenteral antimicrobial therapy [2,3,4,5]. Its spectrum of activity includes Gram-positive microorganisms, including methicillin-resistant *Staphylococcus aureus* (MRSA), coagulase-negative staphylococci (CoNS), β-hemolytic streptococci, *Streptococcus anginosus*, and vancomycin-susceptible *Enteroccoccus faecalis* and *faecium* [2,6].

Multiple studies have shown that dalbavancin has an excellent safety profile and is well-tolerated, being less likely to induce adverse effects, nephrotoxic damage, and drug–drug interactions when administered with other medications, in comparison to other antimicrobials used for the same types of infection [2,5,6,7,8,9,10,11]. Furthermore, its use has been associated with a reduced duration of hospital stays and reduced hospital costs [2,6,11,12], both in outpatient parenteral antibiotic therapy (OPAT) candidates as well as in patients that would conventionally not be eligible for OPAT, like those with unstable medical conditions or intravenous drug use (IVDU) [5,10,13,14,15]. In addition, the use of dalbavancin was associated with a reduced 90-day infection-related readmission (IRR) rate [11]. The effectiveness and safety of dalbavancin extend to the elderly population as well, for which it could constitute a promising therapeutic alternative for the management of deep and complex infections [16].

Dalbavancin has received approval from the Food and Drug Administration (FDA) for use in acute bacterial skin and skin structure infections (ABSSSIs), including erysipelas, cellulitis, major cutaneous accesses, and surgical/traumatic wound infections, caused by Gram-positive cocci like *S. aureus*, *S. anginosus*, *Streptococcus dysgalactiae*, *Streptococcus pyogenes*, *Streptococcus agalactiae*, and vancomycin-susceptible *Enterococcus faecalis*, in both children and adults [2,6,17,18]. However, its pharmacological properties would make it ideal for treatment of other infections as well, such as osteomyelitis, septic arthritis, or even deeper and more serious infections, such as bacteremia, endocarditis, and vascular infections. For example, the use of dalbavancin for treatment of bone and joint infections is supported by the finding that local antibiotic concentrations greater than the MIC90 for *S. aureus* can be sustained for an extended amount of time, implying a significant capacity of tissue penetration and activity against staphylococcal biofilms [19]. Although the question of whether there is a benefit in combining dalbavancin with other antibiotics has not yet been elucidated, Cacopardo et al. suggest that pairing dalbavancin with synergistic drugs suitable for each patient individually might be useful for specific infection categories, including bone and prosthetic joint infections (PJIs) or intravascular infections with no possibility of foreign material removal [20]. Dalbavancin use in off-label indications of infections by Gram-positive microorganisms has yielded promising results [5,21,22,23,24]. The long list of treatment uses of dalbavancin includes consolidation therapy in acute infections [25], catheter-related staphylococcal bacteremia and Gram-positive bacteremia in general [21], infective endocarditis, osteomyelitis, spondylodiscitis, acute septic arthritis, diabetic foot infections, PJIs, suppressive treatment of chronic infections, and pneumonia [5]. Regarding catheter-related staphylococcal bacteremia, dalbavancin was found to be more effective than vancomycin and a factor in reducing treatment cost, length of hospital stay, and need for indwelling devices of parenteral therapy [5,7,12,25]. Many studies have shown that dalbavancin is a safe and effective antibiotic option for osteomyelitis, with cure rates ranging from 65 to 100% [5,6,12,17,18,21,26,27,28,29]. In PJIs, the use of the new compound has shown a cost-saving effect with OPAT, improved patient adherence to therapy, limitation of adverse effects, and a variable cure rate between 33 and 93% [6,9,12,21,27,30,31,32]. Due to its long half-life, this drug may allow treatment of osteoarticular infections, that require prolonged intravenous therapy, with only one or two doses. For example, in osteomyelitis, where the indicated duration of treatment is classically six weeks, dalbavancin has been used in a two-dose schedule that allows minimal patient discomfort and significantly reduces hospital costs [13,28].

The use of dalbavancin has also been explored in relation to the COVID-19 era. The new infection control and hospital resource preservation requirements that emerged due to the pandemic are partially met by the fact that dalbavancin has an exceptionally long half-life that allows for its use in patients in OPAT programs, thus, reducing the spread of nosocomial infections due to a shortened length of a hospital stay [13,33,34,35].

The primary aim of this study was to investigate the clinical and microbiological characteristics of patients to whom dalbavancin was administered in the University Hospital of Heraklion. A secondary aim was to evaluate the use of dalbavancin in regard to the COVID-19 pandemic.

## 2. Materials and Methods

### 2.1. Study Type and Ethics Approval

This is a retrospective single-center study including data regarding all cases of patients treated with dalbavancin in the University Hospital of Heraklion, Heraklion, Greece, a tertiary hospital with 771 beds, from the first time it was administered in 2017 until December 2022. All data were retrieved, retrospectively, using dalbavancin consumption from the pharmacy department, and the patient’s data from the hospital’s electronic medical record and the hard copies of the patients’ files. Data collected and evaluated included microorganisms identified from cultures and antimicrobial resistance of the isolated microorganisms, epidemiological data including patients’ age and gender, their medical history, the number of dalbavancin doses and its dosing scheme, the indication for using dalbavancin, other antimicrobials that may have been used concomitantly to dalbavancin, the outcome of infection (cure or relapse) and patients’ outcome (mortality or survival). Empirical use of dalbavancin implies that the drug was used without available results from microbiology. Post-COVID-19 era was defined as the era from 2020 until the end of the study.

The study was conducted in accordance with the Declaration of Helsinki and approved by the Institutional Review Board of the University Hospital of Heraklion (protocol code 08/24-03-2021).

### 2.2. Statistics

Qualitative data were presented as counts and percentages. Categorical data were analyzed with Fisher’s exact test. Continuous variables were compared using Student’s *t*-test for normally distributed variables and the Mann–Whitney U-test for non-normally distributed variables. For the comparison of more than two continuous variables, the one-way ANOVA test was used for normally distributed variables and the Kruskal–Wallis test for non-normally distributed variables. All tests were two-tailed and a *p*-value of 0.05 or less was considered to be significant. Data are presented as numbers (%) for categorical variables and medians [interquartile range (IQR)] or means (standard deviation (SD)) for continuous variables. A linear-regression analysis model was developed to evaluate the effect of several parameters (age, gender, use before or after the COVID-19 pandemic, empirical dalbavancin use, past medical history, number of dalbavancin doses, indication for dalbavancin use, infection by *S. aureus* or *Enterococcus*, and polymicrobial infection) with cure of infection. All the parameters mentioned above were calculated with GraphPad Prism 6.0 (GraphPad Software, Inc., San Diego, CA, USA). A multivariate logistic regression analysis model was developed to evaluate the association of factors identified in the univariate analysis with a *p*-value < 0.1 with mortality. The multivariate analysis was performed using SPSS version 23.0 (IBM Corp., Armonk, NY, USA).

## 3. Results

In total, 146 patients were included in the study. The median age was 68 years (range: 21–96 years), and 86 (58.9%) patients were male. The medical history of the patients included diabetes mellitus in 44 (33.3%), coronary artery disease in 22 (16.7%), active malignancy in 19 (14.3%), heart failure in 14 (10.6%), chronic kidney disease in 10 (7.6%), and cerebrovascular disease in 6 (4.5%). Moreover, prosthetic orthopedic material was present in 25 (19.2%) of the patients, a prosthetic heart valve in 5 (3.8%), and previous endocarditis in 4 (3.1%). The most common indications for dalbavancin use were osteoarticular infections, followed by ABSSSIs and cardiovascular infections. In 18 (12.3%) patients the infection was not cured, but no patients died. Table 1 shows the characteristics of patients included in the present study in total and in regard to the clinical outcome. Table 2 shows the characteristics of the study’s participants in regard to the indication for dalbavancin use. Empirical use of dalbavancin was most common among patients with ABSSSI, with a frequency of 42.3%. *S. aureus* was identified as a cause for the infection most commonly among patients with cardiovascular infections. Cure rates were different among patients with ABSSSIs, osteoarticular infections, and cardiovascular infections. Among those with ABSSSIs, 95.7% were cured, while the corresponding rates for endocarditis and vascular infections, bacteremia, and osteoarticular infections, were 92.3%, 100%, and 76.4%, respectively. Figure 1a shows the cure rates among the most common indications for dalbavancin use in the present study. Figure 1b shows the microbiology of the most common infections for whom dalbavancin was used in the present study.

Patients who were not cured (had a relapse of the infection) had similar gender, age, medical history, microbiological characteristics, and rate of surgery along with antimicrobial use with those that were cured. However, patients who relapsed were more likely to have an osteoarticular infection (osteomyelitis, prosthetic joint infection (PJI), or septic arthritis) and were less likely to have an ABSSSI as the indication for dalbavancin use.

A comparison of patients treated with dalbavancin in the pre-COVID-19 era with those treated in the post-COVID-19 era revealed that those treated in the pre-COVID-19 era were younger, more likely to have active malignancy, and had received more doses of dalbavancin. There was a non-statistically significant trend for more frequent empirical use of dalbavancin in patients treated in the post-COVID-19 era. Table 3 shows the characteristics of patients treated with dalbavancin in regard to the period they were treated. Figure 1c shows the rates of empirical and targeted dalbavancin use among patients treated before and after the onset of the COVID-19 pandemic in the present study.

In an attempt to identify factors independently associated with a clinical cure, a regression analysis was performed. More specifically, epidemiological characteristics, medical history, clinical characteristics, indications for dalbavancin use, use of other antimicrobials along with dalbavancin, and indication for dalbavancin use were used in a linear regression analysis model to find associates with a cure. The results showed that ABSSSIs were positively associated with a cure and osteoarticular infections were negatively associated with a cure. However, multivariate logistic regression analysis did not identify any factor to be independently associated with a cure. The results of the regression analysis can be seen in Table 4.

## 4. Discussion

The current study presents the characteristics of dalbavancin use in a tertiary hospital from the first time the medication was available in 2017, before the onset of the COVID-19 pandemic, until December 2022. The most common indications for dalbavancin use were osteoarticular infections, followed by ABSSSIs and cardiovascular infections. In 18 (12.3%) patients the infection was not cured, but no patients died. Patients who relapsed were more likely to have an osteoarticular infection and were less likely to have ABSSSI as the indication for dalbavancin use. There was a non-statistically significant trend for more frequent empirical use of dalbavancin in patients treated in the post-COVID-19 era.

Dalbavancin has gained attention during the COVID-19 pandemic for two reasons. First of all, and most importantly, its potential to reduce hospitalizations and minimize patient encounters during the pandemic for the treatment of infections in otherwise stable patients implies that the encouragement of its use could reduce unnecessary risk of COVID-19 transmission from and to patients with an infection by Gram-positive microorganisms that requires treatment. To that end, there are studies suggesting that its use during the COVID-19 pandemic may have increased, allowing the preservation of hospital resources and limiting healthcare exposure [13]. Secondly, dalbavancin gained attention during the pandemic after its ability to bind to ACE2, the receptor where severe acute respiratory syndrome coronavirus-2 (SARS-CoV-2) binds in human cells, was identified [36]. In the same study, dalbavancin administration led to significant inhibition of viral replication and histopathological damage caused by SARS-CoV-2 infections in both rhesus macaque and mouse models. This implied that this medication could have a role in the treatment of patients with COVID-19, even though it was not approved for human use, nor was it studied for this indication.

The present study included data from all patients treated with dalbavancin from the first time it was used in the hospital before the COVID-19 pandemic through the end of 2022, the third year of the pandemic. The profile of patients treated with dalbavancin before the pandemic differs from the one of patients that were treated with it during the pandemic. Patients treated during the pandemic were older and were less likely to have an active malignancy, while the number of doses was lower in this patient population. Furthermore, there was a non-statistically significant trend toward a higher rate of dalbavancin use as an empirical treatment. The lower number of doses used in patients during the COVID-19 pandemic could imply that the drug is more likely to have been used in the context of the continuation of previously active treatment for Gram-positive pathogens, thus facilitating the patient’s discharge. Indeed, a recent study showed that dalbavancin has been successfully used in this context when given some days before discharge to hospitalized patients who are not candidates for OPAT to shorten their hospitalization [37]. In another study with 50 patients with various infections, including ABSSSIs, cardiovascular and osteoarticular infections, the use of dalbavancin after prior treatment with intravenous antibiotics in the hospital was associated with a shorter duration of hospital stay and treatment-related healthcare costs, especially in difficult infections by Gram-positive microorganisms that required prolonged therapy [38]. More specifically, the infections in that study involved ABSSSIs in 20 patients, osteoarticular infections in 18 patients, and vascular graft infections or infections of cardiovascular implantable electronic devices in 12 patients. Patients were treated for a median of 14 days in the hospital with antimicrobials and were then switched to dalbavancin 1500 mg every 14 days until resolution of the infection. A clinical cure was noted in 49 patients (98%), and 37 patients remained without relapse for a median follow-up period of 150 days. This was associated with an estimated cost reduction of EUR 8259 [39].

In our study, the cure rates for patients with ABSSSIs, endocarditis, vascular infections, and bacteremia were over 90%. The high cure rate for ABSSSIs was reasonable and anticipated, given that this constitutes the only indication for the administration of this antibiotic until now [40]. Indeed, this antibiotic has been shown to be effective in other studies, including systematic reviews [38,41]. Importantly, the cure rates for bacteremia, endocarditis, and vascular infections were very high. Notably, the cure rate for endocarditis and cardiac-device-related infections in a recent systematic review of long-acting lipoglycopeptides, including oritavancin and dalbavancin, was 68%, a rate considerably lower than the one noted in this study [42]. In the same systematic review, the cure rate in intravascular-catheter-related infections was 75%, while in the present study, the cure rate for bacteremia was 100%; however, the data referred to bacteremia in general, not only to intravascular-catheter-related infections [42]. The successful use of dalbavancin for these infections, even though not yet approved for their treatment, reinforces the need for conduction of high-quality randomized trials that will formally assess its safety and effectiveness for the aforementioned conditions. This information could lead to the approval of this drug, allowing its widespread use that could reduce hospital costs and increase patient satisfaction for infections that typically require very prolonged treatment, such as infective endocarditis.

The cure rate for patients with osteoarticular infections was about 75% in our study. Even though this rate is not as high as in the previously mentioned infections, it is in line with the data from a systematic review with data from 14 studies, where the cure rate for these infections was 73% [42]. However, Rappo et al., Almangour et al., and Wunsch et al. reported cure rates of approximately 90% for dalbavancin—the highest rates for this type of infection [27,28,29]. However, the studies by Rappo et al. and Almangour et al. described patients with osteomyelitis only. Furthermore, in the aforementioned studies, surgery along with antibiotic use was performed in more than 80% of patients, while in this study, the corresponding rate was much lower.

In this study, the most commonly identified pathogens were coagulase-negative staphylococci. However, in the case of endocarditis and vascular infections, *S. aureus* was the most commonly identified microorganism. This is in line with some studies where coagulase-negative microorganisms were also the most frequently isolated pathogens in the case of osteoarticular infections [9,43]. However, in other studies, *S. aureus* was the most commonly identified pathogen [28,29]. Interestingly, in many such studies, *S. aureus* was methicillin-sensitive, implying that dalbavancin may have been used not due to a lack of antimicrobial options, but rather for its potential to reduce hospitalization, decrease healthcare costs, and increase patient convenience [28,29]. In the case of cardiovascular infections, *S. aureus* was the most commonly identified pathogen in several studies, followed by coagulase-negative staphylococci, as was the case in our cohort [44,45].

This study has some notable limitations, such as its retrospective design. Due to the retrospective design, there were limited missing values in the current data. Moreover, the data are derived from a single tertiary hospital, thus the interpretation of the results should be performed with caution.

## 5. Conclusions

The present study examines dalbavancin use in a tertiary hospital from the first time it was used before the COVID-19 pandemic, in 2017, up to the end of 2022. The most common indications for dalbavancin use were osteoarticular infections, followed by ABSSSIs, and cardiovascular infections. The antibiotic was used empirically in one out of three cases, most commonly due to ABSSSIs and most commonly in the post-COVID-19 era. Dalbavancin use was more common in the post-COVID-19 era, and patients’ characteristics had some differences, such as patients’ age, which was higher in the post-COVID-19 era. In 12.3%, the infection was not cured, but no patients died. For patients with ABSSSIs, endocarditis and vascular infections, and bacteremia, the cure rates were more than 90%, and in osteoarticular infections, the cure rate was 76.4%. Thus, dalbavancin has great potential for use in complicated and invasive infections that may require prolonged intravenous antimicrobial treatment. However, since it is currently approved only for ABSSSIs, further studies are required to formally investigate its role in such infections, such as endocarditis, vascular infections, and osteoarticular infections.

## Figures and Tables

**Figure 1 antibiotics-12-01205-f001:**
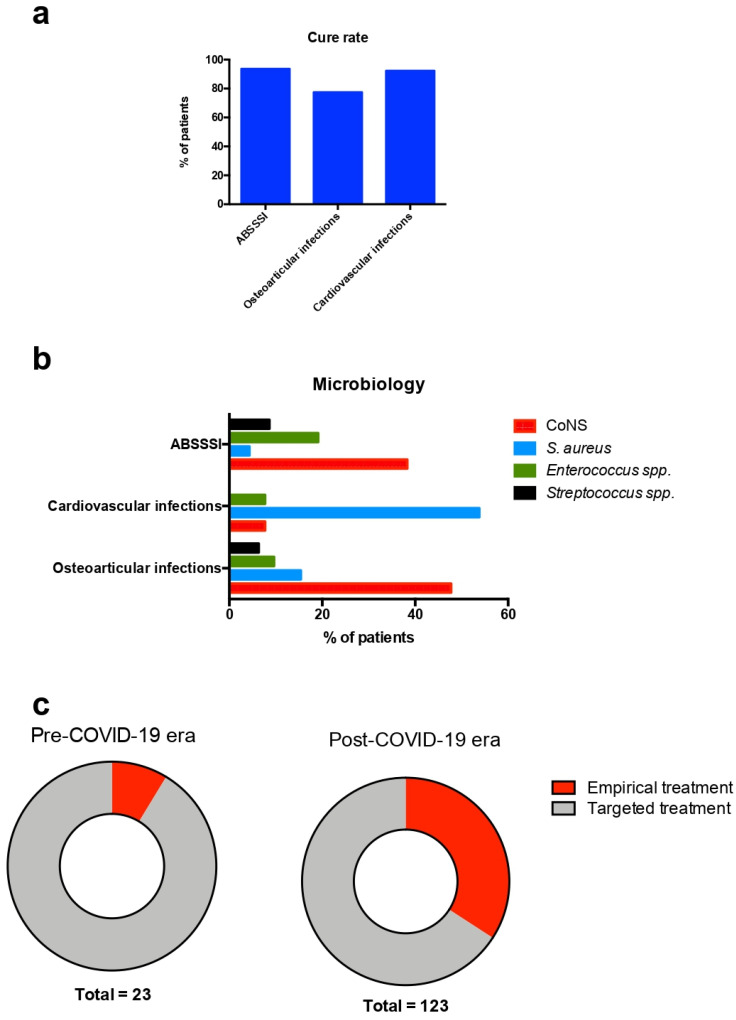
Cure rates, microbiology, and empirical use of dalbavancin in the present study. (**a**) Cure rates for the most common indications for dalbavancin use, (**b**) microbiology of the most common conditions for whom dalbavancin was used, and (**c**) rates of empirical and targeted dalbavancin use in regard to the era dalbavancin was used. ABSSSI: acute bacterial skin and skin structure infection; CoNS: coagulase-negative *Staphylococcus*; and COVID-19: coronavirus disease 2019.

**Table 1 antibiotics-12-01205-t001:** Characteristics of the study’s participants in total and in regard to the clinical outcome.

Characteristic	All Patients (*n* = 146)	Infection Cured (*n* = 128)	Infection Not Cured (*n* = 18)	*p*-Value
Age, years, median (IQR)	68 (56–78)	67.5 (55–78.3)	67 (56–78)	0.7029
Male, *n* (%)	86 (58.9)	76 (59.4)	10 (55.6)	0.8015
Medical history				
Prosthetic orthopedic material, *n* (%)	25 (19.2)	22 (19.5)	3 (17.6)	1
Active malignancy, *n* (%)	19 (14.3)	18 (15.5)	1 (5.9)	0.4650
Chronic kidney disease, *n* (%)	10 (7.6)	9 (7.8)	1 (5.9)	1
Diagnosis for dalbavancin use				
Osteomyelitis, PJI or septic arthritis, *n* (%)	55 (43)	42 (37.8)	13 (76.5)	0.0036
ABSSSI, *n* (%)	47 (36.7)	45 (40.5)	2 (11.8)	0.0289
Endocarditis or vascular graft infection, *n* (%)	13 (10.2)	12 (10.8)	1 (5.9)	1
Bacteremia *, *n* (%)	6 (4.7)	6 (5.4)	0 (0)	1
Empirical treatment, *n* (%)	44 (32.4)	40 (33.9)	4 (22.2)	0.4225
Polymicrobial, *n* (%)	45 (33.1)	39 (33.1)	6 (33.3)	1
Microbial isolation				
Coagulase negative-staphylococci, *n* (%)	64 (69.6)	56 (71.8)	8 (57.1)	0.3457
*Staphylococcus aureus*, *n* (%)	25 (27.2)	20 (25.6)	5 (35.7)	0.5164
*Enterococcus* spp., *n* (%)	20 (21.7)	17 (21.8)	3 (21.4)	1
*Streptococcus* spp., *n* (%)	7 (7.6)	5 (6.4)	2 (14.3)	0.2880
Number of dalbavancin doses, median (IQR)	2 (1–3)	1 (1–2.5)	2 (1–3)	0.3244
Surgical management, *n* (%)	42 (32.3)	36 (31.9)	6 (35.3)	0.7856
Mortality, *n* (%)	0 (0)	0 (0)	0 (0)	NA

ABSSSI: acute bacterial skin and skin structure infection; IQR: interquartile range; NA: not applicable; and PJI: prosthetic joint infection. * Bacteremia without endocarditis or vascular graft infection.

**Table 2 antibiotics-12-01205-t002:** Characteristics of the study’s participants in regard to the indication for dalbavancin use.

Characteristic	ABSSSIs (*n* = 47)	Osteoarticular Infections (*n* = 55)	Endocarditis and Vascular Infections (*n* = 13)	*p*-Value
Age, years, median (IQR)	66 (54–77)	74 (59.5–78)	76 (43.5–81.5)	0.4529
Male, *n* (%)	27 (57.4)	27 (50.9)	10 (76.9)	0.2357
Medical history				
Prosthetic orthopedic material, *n* (%)	8 (17.4)	16 (32)	1 (8.3)	0.1033
Active malignancy, *n* (%)	8 (17)	3 (6)	1 (8.3)	0.2120
Chronic kidney disease, *n* (%)	5 (10.6)	3 (6)	1 (8.3)	0.7089
Empirical treatment, *n* (%)	20 (42.3)	16 (30.8)	2 (15.4)	0.1507
Polymicrobial, *n* (%)	15 (31.9)	16 (30.8)	5 (38.5)	0.8676
Microbial isolation				
Coagulase negative-staphylococci, *n* (%)	18 (38.3)	21 (47.7)	1 (7.7)	0.0934
*Staphylococcus aureus*, *n* (%)	2 (4.3)	8 (15.4)	7 (53.8)	<0.0001
* Enterococcus* spp., *n* (%)	9 (19.1)	5 (9.6)	1 (7.7)	0.2684
* Streptococcus* spp., *n* (%)	4 (8.6)	3 (6.3)	0 (0)	0.5056
Number of dalbavancin doses, median (IQR)	1 (1–2)	2 (1–4)	1 (1–2)	0.0776
Surgical management, *n* (%)	9 (19.6)	19 (36.5)	5 (41.7)	0.1637
Clinical cure, *n* (%)	45 (95.7)	42 (76.4)	12 (92.3)	0.0148
Mortality, *n* (%)	0 (0)	0 (0)	0 (0)	NA

ABSSSI: acute bacterial skin and skin structure infection; IQR: interquartile range; and NA: not applicable.

**Table 3 antibiotics-12-01205-t003:** Characteristics of the study’s participants in regard to the period they were treated.

Characteristic	Pre-COVID-19 Era (*n* = 23)	Post-COVID-19 Era (*n* = 123)	*p*-Value
Age, years, median (IQR)	56 (50–71)	69.5 (57–80)	0.0061
Male, *n* (%)	12 (52.2)	74 (60.2)	0.4964
Medical history			
Prosthetic orthopedic material, *n* (%)	4 (25)	21 (18.4)	0.5092
Active malignancy, *n* (%)	6 (35.3)	13 (11.2)	0.0174
Chronic kidney disease, *n* (%)	0 (0)	10 (8.7)	0.3591
Diagnosis for dalbavancin use			
ABSSSI, *n* (%)	5 (29.4)	42 (37.8)	0.5962
Osteomyelitis, PJI or septic arthritis, *n* (%)	10 (58.8)	45 (40.5)	0.1920
Endocarditis or vascular graft infection, *n* (%)	0 (0)	13 (11.7)	0.2138
Bacteremia *, *n* (%)	0 (0)	6 (5.4)	1
Empirical treatment, *n* (%)	2 (11.1)	42 (35.6)	0.0559
Polymicrobial, *n* (%)	6 (33.3)	39 (33.1)	1
Microbial isolation			
Coagulase negative-staphylococci, *n* (%)	9 (56.3)	55 (72.4)	0.2375
*Staphylococcus aureus*, *n* (%)	7 (43.8)	18 (23.7)	0.1254
*Enterococcus* spp., *n* (%)	4 (25)	16 (21.1)	0.7436
*Streptococcus* spp., *n* (%)	0 (0)	7 (9.2)	0.3477
Number of dalbavancin doses, median (IQR)	2 (1–5)	1 (1–2)	0.0018
Surgical management, *n* (%)	4 (28.6)	38 (32.8)	1
Clinical cure, *n* (%)	19 (82.6)	109 (88.6)	0.4875
Mortality, *n* (%)	0 (0)	0 (0)	NA

ABSSSI: acute bacterial skin and skin structure infection; COVID-19: coronavirus disease 2019; IQR: interquartile range; NA: not applicable; and PJI: prosthetic joint infection. * Bacteremia without endocarditis or vascular graft infection.

**Table 4 antibiotics-12-01205-t004:** Regression analysis of clinical cure.

Parameter	Univariate Analysis *p*-Value	Multivariate Analysis *p*-Value	OR (95% CI)
ABSSSIs	0.081	0.866	1.173 (0.184–7.502)
Osteomyelitis, PJI, or septic arthritis	0.0106	0.114	0.28 (0.058–1.355)

ABSSSI: acute bacterial skin and skin structure infection; CI: confidence intervals; PJI: prosthetic joint infection; and OR: odds ratio.

## Data Availability

The data presented in this study are available on request from the corresponding authors.

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
