# Peer review of "Real-World Data Regarding Dalbavancin Use before and during the COVID-19 Pandemic—A Single-Center Retrospective Study"

_antibiotics, 2023, doi:10.3390/antibiotics12071205_

Round 1

Reviewer 1 Report

The manuscript entitled "Real-world data regarding dalbavancin use before and during the COVID-19 pandemic - a single-centre retrospective study", is original. To improve the study, I recommend:

line 115: Please provide the number of approvement.

line 149, 153, 168: Tables 1, 2 and 3 are difficult to follow in the "Characteristic" column. I recommend changing the centred text alignment to a different format that is easier to follow.

line 153: The data in Table 2 requires statistical analysis.

lines 194 - 209: In the "Discussion" chapter, the information about Dalbavancin should be placed in the "Introduction" chapter.

lines 235-238: Multiple studies have been referenced in this case, so it is necessary to present them as bibliographic citations.

lines 238-242: It is necessary to provide a brief presentation of the referenced study.

Author Response

The manuscript entitled "Real-world data regarding dalbavancin use before and during the COVID-19 pandemic - a single-centre retrospective study", is original. To improve the study, I recommend:

line 115: Please provide the number of approvement.

Response: Thanks. This information was provided at the end of the manuscript, as usual for the journal Antibiotics. However, following the reviewer’s comment, we added that information at that specific point at the end of 2.1 subsection, as can be seen in the revised version of the manuscript.

line 149, 153, 168: Tables 1, 2 and 3 are difficult to follow in the "Characteristic" column. I recommend changing the centred text alignment to a different format that is easier to follow.

Response: We changed the format of the tables as well as the alignment of the first column, as suggested by the reviewer. Now, the tables are easier for the reader to understand, as can be seen in the revised version of the manuscript.

line 153: The data in Table 2 requires statistical analysis.

Response: Thanks for the comment. We added statistical analysis in that table and also added a description of how that was done in the subsection 2.2 in the methods section. Based on the results, clinical cure was significantly different among the three major clinical entities in this study. Moreover, there was a statistically significant difference in the rate of isolation of S. aureus. These changes can be seen in the revised version of the manuscript.

lines 194 - 209: In the "Discussion" chapter, the information about Dalbavancin should be placed in the "Introduction" chapter.

Response: Thanks. We have moved that paragraph in the introduction section. To make it better for the reader, the paragraph was segmented and sentences were put in different parts of the introduction section, while, some sentences were deleted. This can be seen in the introduction section of the revised manuscript.

lines 235-238: Multiple studies have been referenced in this case, so it is necessary to present them as bibliographic citations.

Response: Sorry, that was our mistake. We meant ‘a recent study’. The use of plural was wrong. The reference that was cited was the one intended for bibliographic citation.

lines 238-242: It is necessary to provide a brief presentation of the referenced study.

Response: Thanks. We have added a description of that study right after the sentence highlighted by the reviewer to allow the reader to understand the rationale of the study and the hospital cost that was saved due to the switch to dalbavancin. This can be seen in the discussion section of the revised manuscript.

Reviewer 2 Report

The article's structure was well-organized, making it easy to follow along. It began with a compelling introduction that immediately grabbed my attention, posing questions and introducing a fascinating topic.

As a minor, I appreciated their ability to make complex concepts accessible and relatable to readers of all ages.

Please reformulate paragraph 115-117.

As a minor, I often find that visual aids enhance my understanding and make the content more engaging. Integrating relevant images, graphs, or diagrams would have further elevated the article's appeal.

While there was room for improvement in terms of visual aids, the article's overall impact on me as a minor reader was undeniably positive. I eagerly look forward to discovering more thought-provoking works by this talented author in the future.

The article's structure was well-organized, making it easy to follow along. It began with a compelling introduction that immediately grabbed my attention, posing questions and introducing a fascinating topic.

As a minor, I appreciated their ability to make complex concepts accessible and relatable to readers of all ages.

Please reformulate paragraph 115-117.

As a minor, I often find that visual aids enhance my understanding and make the content more engaging. Integrating relevant images, graphs, or diagrams would have further elevated the article's appeal.

While there was room for improvement in terms of visual aids, the article's overall impact on me as a minor reader was undeniably positive. I eagerly look forward to discovering more thought-provoking works by this talented author in the future.

Author Response

The article's structure was well-organized, making it easy to follow along. It began with a compelling introduction that immediately grabbed my attention, posing questions and introducing a fascinating topic.

Response: Thanks for the comment

As a minor, I appreciated their ability to make complex concepts accessible and relatable to readers of all ages.

Response: Thanks

Please reformulate paragraph 115-117.

Response: Thanks. We changed that paragraph by adding some more information, as well as the information on the protocol code and the date of approval, as suggested by another reviewer.

As a minor, I often find that visual aids enhance my understanding and make the content more engaging. Integrating relevant images, graphs, or diagrams would have further elevated the article's appeal.

Response: Thanks. We added a figure, figure 1, that we feel that summarizes the main findings of the study in an informative way. We feel that the reader will draw important conclusions from this graph, and will help the manuscript better convey its basic message.

While there was room for improvement in terms of visual aids, the article's overall impact on me as a minor reader was undeniably positive. I eagerly look forward to discovering more thought-provoking works by this talented author in the future.

Response: Thanks.

Reviewer 3 Report

Having evaluated the work presented, I believe that it does not add anything new to what has been published in the literature. Communicating the experience of a single centre does not seem sufficient to consider publishing the article in the journal.  Therefore I would not consider the article for publication in the journal. 

No commets on the  quality of English Language. 

Author Response

Having evaluated the work presented, I believe that it does not add anything new to what has been published in the literature. Communicating the experience of a single centre does not seem sufficient to consider publishing the article in the journal.  Therefore I would not consider the article for publication in the journal.

Response: Thanks for the comment. Even though we do agree that there are several published studies regarding dalbavancin use, we are skeptical about this comment. First of all, we believe that adding information regarding experience with off-label use of a drug that has not yet been formally approved for these indications is important for clinicians. Since this drug is not approved for these uses, the only way to collect data on success and failures is through scientific research. Thus, even if this study included a repetition of other studies’ results, it would still have been relatively valuable. Moreover, to our knowledge, the studies providing information on off-label dalbavancin use with a temporal analysis (as in this case, where the data are presented in regard to the onset of the COVID-19 pandemic) are quite few, and the present study adds interesting information from that perspective and is novel in that way.

Round 2

Reviewer 1 Report

Thank you for the responses. The article does not require any adjustments.

Reviewer 3 Report

After reviewing the article submitted for review, I believe that the deficiencies detected and which did not recommend publication in the journal persist, so I would not recommend publication in the journal.